# Parkinson’s Disease: A Prionopathy?

**DOI:** 10.3390/ijms22158022

**Published:** 2021-07-27

**Authors:** Sarah Vascellari, Aldo Manzin

**Affiliations:** Microbiology and Virology Unit, Department of Biomedical Sciences, University of Cagliari, 09042 Cagliari, Italy; aldo.manzin@gmail.com

**Keywords:** Parkinson’s disease, α-synuclein, prion-like mechanism, protein misfolding co-factors, gut microbiota

## Abstract

The principal pathogenic event in Parkinson’s disease is characterized by the conformational change of α-synuclein, which form pathological aggregates of misfolded proteins, and then accumulate in intraneuronal inclusions causing dopaminergic neuronal loss in specific brain regions. Over the last few years, a revolutionary theory has correlated Parkinson’s disease and other neurological disorders with a shared mechanism, which determines α-synuclein aggregates and progresses in the host in a prion-like manner. In this review, the main characteristics shared between α-synuclein and prion protein are compared and the cofactors that influence the remodeling of native protein structures and pathogenetic mechanisms underlying neurodegeneration are discussed.

## 1. Introduction

Parkinson’s disease (PD) is the most common movement disorder that has been first described by James Parkinson in 1817. Mentioning his own work, he defined this pathology as “the shaking palsy”, with patients having “involuntary tremulous motion, with a propensity to bend the trunk forward and to pass from a walking to a running pace, while senses and intellects seem intact” [1]. In addition to Parkinson’s description, it has been attested that the disease most frequently induces rigidity and slowness of movements, or “akinesia”, which together with tremors, constitute the so-called “triad” of cardinal motor symptoms [2,3]. The motor signs are caused by a reduction of striatal dopamine resulting by the progressive and severe loss of dopaminergic neurons in the substantia nigra pars compacta [4]. The most relevant mechanisms involved in the dopaminergic neuronal loss include the accumulation of intraneuronal protein inclusions, known as Lewy bodies, which are composed mainly of misfolded α-synuclein aggregates [5]. Although the causes of these pathological changes are still unclear, genetic and non-genetic factors, aging, exposure to environmental components, and oxidative stress may be involved. A recent striking theory links several neurodegenerative proteinopathies, referring to the conformational change of a protein as a common disease-causing mechanism. This theory emphasizes the “prion-like” activity of α-synuclein, which shows ability of self-aggregation and self-cell to cell propagation. In this review, the controversial question of whether PD can be considered a “prionopathy” is discussed, summarizing the main evidence supporting or countering this view.

## 2. α-Synuclein and Lewy Pathology

The α-synuclein is a neuronal cytosolic protein, first described by Maroteaux et al. in 1988: he reported that this protein is localized as soluble or membrane-associated fraction in the nucleus and presynaptic nerve terminals [6,7]. However, it was in 1997 that interest in the involvement of α-synuclein in the pathogenesis of PD was renewed and such involvement was identified as a mutation in the gene associated with PD cases [7]; moreover, α-synuclein deposits as the main protein component of Lewy bodies and Lewy neurites became the hallmark of the disease [8]. α-synuclein belongs to the synuclein family, members of intrinsically disordered proteins, also known as natively unfolded proteins, which are characterized by low hydrophobicity and high net charge [9]. This monomeric unfolded protein, encoded by *SNCA* gene, is composed of 140 amino acids and its structure can be categorized in three domains, whereas C-terminal end contains the sites for interaction with other proteins and molecules, the first two domains, N-amino terminus and a non-amyloid component, include a membrane-binding domain [10]. Progressively obtained data show that these domains are crucial for the pathogenic process underlying PD.

## 3. The Prion Concept

To date, the innovative concept of prion as “proteinaceous infectious particle” is universally accepted by most scientists in the field [11,12]. In 1997, when evidence of the link between α-synuclein and PD was officially recognized, Stanley Prusiner was awarded the Nobel Prize for his pioneering work on prion disease, in which he presented the revolutionary theory according to which prions are “infectious agents able to replicate and propagate in the host in the absence of nucleic acids and endowed of strain specific properties” [11,12]. What was known about these new agents at the time was that they played a pathogenic role in a group of non-ordinary neurodegenerative disorders, known as Transmissible Spongiform Encephalopathies (TSEs) due to their transmissible nature. TSEs affect animals, such as Scrapie and Bovine Spongiform Encephalopathy (BSE), and humans, such as Creutzfeldt–Jakob disease (CJD) and its “new variant” (vCJD), and Kuru which remains one of the most interesting and enigmatic stories of human-to-human transmission with relations to cannibalistic practices. Except for the iatrogenic forms, due to transmission through contaminated tissues, and the sporadic one, in which an infectious origin has not yet been demonstrated, these neurodegenerative diseases are examples of zoonosis, and are attributable to the phenomenon known as “spillover” from animals to humans (e.g., vCJD, from cow to human), or from animals to other animals (e.g., BSE, from sheep to cow). The concept of transmission of a protein as a pathogen that unconventionally associated one species with another was accepted for the first time. Neurodegenerative proteinopathies (e.g., PD) other than true prion diseases share similar pathogenetic features, although each is characterized by distinctive clinical features:

### 3.1. First, the Protein Structure

The prion protein exists in different conformational states. Two isoforms of prion proteins are known: the physiological one, encoded by PRNP gene, which is the cellular prion protein (PrP^C^) with a the predominantly α-helical shape, and its pathological misfolded counterpart (PrP^Sc^) rich in β-sheet, which can aggregate and accumulate in different regions of the brain and induce distinct neuropathologies depending on specific prion strain or disease. Similarly to prions, α-synuclein under non-pathological conditions results in an unstable equilibrium toward an unfolded state [13]. On the other hand, under pathological conditions, α-synuclein monomers shift toward a misfolded shape rich in β-sheet, which leads to the self-aggregation of the protein and then it deposits into Lewy bodies. Accordingly, both physiological forms of PrP^C^ and α-synuclein are prone to self-aggregation, increasing the amount of β-pleated sheet forms and therefore changing their physico-chemical and biological characteristics, including the resistance to ordinarily denaturing agents [14].

### 3.2. Second, the Aggregation Pathway 

The more accredited seeding hypothesis on the amplification mechanism of native proteins [15,16] suggests a progression from the clustering of larger oligomers to the formation of protofibrils, fibrils and aggregates. The proposed model predicts that misfolded exogenous forms of PrP^Sc^ interact with endogenous α-helical forms of PrP^C^ triggering their conformational transition into pathological forms. The oligomers are responsible for impaired and neuronal transmission [17] accumulate and recruit other misfolded prion proteins to form aggregates that subsequently break into small fragments which, like seeds, promote the self-aggregation process [18,19]. This model of polymerization of seeded protein underlies the pathological process in PD in which the abnormal α-synuclein recruits physiological α-synuclein to induce its continuous conversion into misfolded forms [20]. Therefore, both normal forms of PrP^C^ and α-synuclein are required as substrates for the self-aggregation and self-propagation processes [20,21,22] that consequently seem to occur in sites particularly rich in native substrates, such as synaptic or cellular sites, following the cellular trafficking of these proteins [23].

Several in vitro studies have given an indisputable proof of this *seeded* protein polymerization model mimicking the pathogenic process. Currently this model is widely applied to detect aggregates of PrP^Sc^ and α-synuclein in several tissues and biological fluids using the innovative assay developed by Caughey’s group [24,25,26] and then adapted by others [27,28,29], known as Real-time quaking-induced conversion (RT-QuIC) assay.

### 3.3. Third, Cell to Cell Transmission

The theory of cell-to-cell transmission of pathological proteins has been restricted exclusively to prions for two decades [23]. In contrast, the first evidence of α-synuclein transmission dates back to 2008, when a couple of seminal papers reported that subjects who suffered from PD developed Lewy α-synuclein pathology within grafted dopaminergic neurons that had been implanted many years earlier [30,31]. Following these findings, many in vitro and in vivo experiments revealed that cell-to-cell transmission of α-synuclein was possible [23,32,33,34].

This process leads to cellular diffusion in order to ensure the effective propagation of said proteins which are released to enter other cells according to a prion-like pattern. Unlike prions, which are mainly located on the outer side of the plasma membrane, intracellular α-synuclein must cross the membrane and be transported into the extracellular space. Commonly, the release of cytosolic α-synuclein into the extracellular space and biological fluids (e.g., blood and CSF) appears to be mediated by damage to neuronal membranes [23,35].

It has been suggested that the different localization of native PrP^C^ and α-synuclein may play a role in the efficacy of the conversion and propagation process of misfolded forms, as said process would be more efficient on the cell surface (PrP^C^) than in the cytosol (α-synuclein) [23].

This could explain the different clinical progression of the human form of prion disease (e.g., CJD), which is characterized by rapidly progressive dementia compared to PD.

Interestingly, it is believed that the transport of prion and α-synuclein seeds outside the cell [23,32,33], and their passage into adjacent cells involves several secretory pathways capable of establishing a cell-to-cell connection, such as endosomal vesicles, exosomes, tunneling nanotubes, synapses, and receptors [20,36,37,38,39]. In this seeding propagation process the misfolded prion and newly internalized α-synuclein can serve as a substrate capable of recruiting new monomers of PrP^C^ and α-synuclein [40,41] and accumulating within cells before being transmitted again.

### 3.4. Fourth, Point of Origin and Host Invasion

Although prion diseases are mainly limited to the central nervous system (CNS), except for sporadic and familial forms, most of them originate in the periphery. Many examples have been reported in the literature of animal prion diseases (e.g., scrapie, BSE, and chronic wasting disease), in which an origin from the oral or nasal cavity and the subsequent retrograde spread to CNS through neural tracts or possibly through blood has been suggested [42,43,44,45]. In humans, vCJD is also believed to originate from ingestion of contaminated meat from animals with BSE. Prions have been suggested to disseminate via the peripheral gastrointestinal tract to the CNS.

Neuroinvasion begins after self-aggregation and self-propagation that initially occurs within the secondary lymphoid organs, where various cells associated with the intestine are involved (e.g., enterocytes, M-cells, dendritic cells, mononuclear phagocytes, and B cells). Then, prions cross the intestinal barrier and spread from enteric nervous systems (ENS), through peripheral nerves (e.g., splanchnic nerve and vagus nerve) into the CNS, where they induce widespread neuronal loss [45,46,47,48]. In support of these theories, prion neuroinvasion from ENS is inhibited after sympathetic nerves depletion [49]. On the contrary, prion dissemination from the CNS to peripheral tissues and organs occurs through anterograde spreading along nerve fibers, peripheral synapses (e.g., neuromuscular junction) peripheral tissues (i.e., muscle cells and mucosa), lymph, and blood [50]. Interestingly, an example of prion dissemination to the peripheral mucosa is documented in the olfactory system. Here, prions spread along synapses within olfactory neurons via the olfactory and vomeronasal cranial nerves and reach olfactory sensory mucosa with subsequent release of prions into the nasal cavity [50,51,52].

Regarding PD, there are conflicting opinions concerning the site of origin of the pathological process. One of the most captivating and controversial hypotheses comes from the post mortem studies of Braak and colleagues dating back to 2003 [53]. In his work, Braak has proposed that α-synuclein pathology may start from enteric nerves in the gut or the olfactory bulb and progressively spread in stages along neural tracts to the brain [53,54]. The authors point out that these two sites are predominantly exposed to exogenous triggers, such as toxins, pesticides, pathogens, viruses, and bacteria, suggesting the concept that “a virus or a prion-like pathogen, consisting of *misfolded* α-synuclein, could be responsible for the staged progression” [55,56].

The neuroinvasion by pathological α-synuclein, which follows that of the vagus nerve which performs autonomic functions, seems to explain the symptoms that often precede motor dysfunctions, such as gastrointestinal and smell disorders and the premotor component of REM sleep behavior disorder (RBD) [56].

Interestingly, pathologic α-synuclein was found in the gastrointestinal tract and olfactory bulb of patients who suffered from PD, as well as in RBD, which is considered prodromal PD [57]. Consistent with “gut–brain” theory, recent studies have demonstrated the oral transmission of a-synuclein fibrils in transgenic mice [58] and the protective role of vagotomy in PD development, suggesting the vagal nerve as a possible route for the spreading of pathogenic α-synuclein [59].

A critical issue in the Braak hypothesis is that the proposed α-synuclein spreading pattern has not been confirmed in some histopathological studies of PD. One work has suggested a reverse path of α-synuclein spreading which could originate from the brain, and then reach the gastrointestinal tract [60].

In this regard, it is interesting to note that a recent work has suggested a hypothesis that encompasses previous theories on intestinal or CNS origin considering both possible. The authors presented a new classification of PD, depending on the different origin of α-synuclein. A “brain-first” subtype, in which α-synuclein pathology originates primarily in the brain, and a “body-first” subtype, characterized by premotor RBD and gastrointestinal impairment, in which the disease begins in the enteric or peripheral autonomic nervous system and then spreads to the brain [61].

Another possible explanation for the reported discrepancies could be that α-synuclein pathology is a feature shared with others Lewy diseases, such as multiple system atrophy (MSA) [62]. Interestingly, several authors have suggested that PD and MSA may be caused by distinct strains of the same pathological α-synuclein, responsible of inducing distinct neuropathologies, as prions do.

Although the Braak hypothesis remains a matter of debate and criticism, it represents a milestone in the pathogenesis studies of PD that links to the prion-like theory.

## 4. Co-Factors Involved in the *Misfolding* Process

Another remarkable point of interest is whether the molecular event, based on protein self-aggregation, involves other co-factors for the development and progression of PD and prion diseases. Several studies have shown that prion and α-synuclein transmission is more efficient when the inoculum of protein aggregates includes tissue extracts, in which other components may act as ”co-factors” [23]. In this regard, it has been proposed that many environmental conditions and intrinsic factors can influence the pathological process. In addition to specific genetic factors and post-translational modifications [63,64], other non-genetic factors are believed to be involved. Among these, the interaction with some lipids, metal ions, pesticides, xenobiotics, changes in the environment such as oxidative stress, toxic insults, inflammation, pH reduction, proteins, and bacterial products are some of the factors that can favor the oligomerization of α-synuclein and PrP^C^. The main factors which can be involved in these processes are summarized as follows:

### 4.1. Gene Mutations

Although most human prion diseases and PD are sporadic, 10–15% of cases are classified as familial, which indicates the critical role of gene mutations in the pathogenesis of these disorders [65]. Mutations in the prion protein gene (PRNP) are associated with inherited forms of GSS (e.g, P102L, P105L, P105S, A117V, F198S, and Q217R), FFI (e.g., D178N), and gCJD (e.g., D178N, V180I, H187R, E200K, R208H, and V210I) with autosomal dominant patterns [66]. While fewer gene mutations represent octapeptide expansions or deletions, most prevalent mutations are point mutations (missense mutations) located in the N- or C-terminal region of the PRNP gene and they show a complete penetrance. On the contrary, E200K and V210I mutations are associated with a variable penetrance depending on age [66,67]. Likewise, most of inherited forms of PD appear to be caused by point mutations (missense mutations) or multiplications (duplication and triplication) of the α-synuclein gene (SNCA) with incomplete penetrance depending on age [65,68]. The pathogenic mutations linked to SNCA gene (e.g., A53T, A53E, A30P, E46K, H50Q, and G51D) occur in the N-terminal region of α-synuclein and are related to an early onset and rapid progression of PD. Conversely, mutations of Leucine-rich repeat kinase 2 (LRRK2), which represent an increased risk for the development of inherited PD, are associated with a late-onset of the disease (e.g., G2019S) [68]. Whereas the predominant pathogenic mutations in both SNCA and LRRK2 result in autosomal dominant pattern similarly to inherited forms of human prion diseases, other mutations (e.g., PARK2, PARK7, and PARK6) that involve PRKN, DJ-1, and PINK1 genes result in autosomal recessive patterns [65].

### 4.2. Post-Translational Modifications

A dysregulation of post-translational modifications (PTMs), due to, for example, inflammation conditions, can alter the structure of susceptible proteins, such as PrP^C^ and α-synuclein, promoting their transition into toxic oligomers, fibrils, and aggregates [64,69,70]. PrP^C^ and α-synuclein undergo PTMs, such as addition of functional groups and proteolytic cleavage (e.g., glycosylation, acetylation, nitration, oxidation, and phosphorylation), during and after their biosynthesis. In particular, the common pathological PTMs of both PrP^C^ and α-synuclein proteins include phosphorylation and nitration. Phosphorylation induce anionic conditions that favor the conformational changes of proteins [64]. While pathological phosphorylation of PrP^C^ mediated by cyclin-dependent kinase 5 involves N-terminal region at amino acid residue Ser-43, the prevalent phosphorylation of α-synuclein involves the C-terminal region at amino acid residue Ser-129 and can be mediated by several kinases (e.g., casein kinase I and II, the G protein-coupled receptor kinases, Leucine-rich repeat kinase 2, and polo-like kinases) [69,70]. These phosphorylated sites are associated with the self-aggregation of proteins and the development of proteinopaties in human prion diseases and PD. Regarding PD, it is indeed known that over 90% of α-synuclein in the Lewy bodies is phosphorylated at Ser-129. Moreover, nitration modifications of tyrosine residues, which are induced by the action of oxygen and nitric oxide and their products, involve multiple residues of both PrP^C^ (Y131, Y148, Y152, Y153, Y158, Y221, Y158, Y221, and Y227/Y228) and α-synuclein (Y125, Y133, and Y136) and occur predominantly in the C-terminal region of the proteins. Several evidences have demonstrated that nitration increases the aggregation of PrP^C^ and the oligomerization of α-synuclein [69,70].

### 4.3. Lipid Interactions

The composition of lipids in membranes is critical for the oligomerization of PrP^C^ and α-synuclein. Whereas PrP^C^ is a glycoprotein attached to the membrane through a glycosyl-phosphatidyl-inositol anchor [71], the N-terminus of α-synuclein can assume α-helical shapes binding to protein or lipid membranes [72]. In vivo and in vitro studies [73,74,75,76] have demonstrated that the lipid component of the membrane, and in particular the cholesterol- and sphingolipid-rich domains, also known as lipid rafts, could modulate the tendency of these protein to change their conformational state. Lipid rafts appear to act as a shuttle, promoting the binding and oligomerization of α-synuclein and PrP^C^ [77].

### 4.4. Metals Ions, Pesticides, and Molecular Crowding

Several metals ions have shown to play a role in the misfolding process [78], including copper. In fact, both α-synuclein and PrP^C^ are copper-binding proteins, and copper is one of the metals reported to interfere with protein folding; its aggregation results in the formation of toxic protein aggregates [79,80]. Copper binds α-synuclein and PrP^C^ in the N-terminal and C-terminal domains, which in PrP^C^ consist of an octarepeat site and a non-octarepeat site with high affinity for this metal [81,82].

Although it is still a matter of discussion whether copper can promote or prevent the aggregation of these proteins, and its exact mode of action, several mechanisms of interaction have been suggested: binding to protein functional groups, removal of metal ions essential in metalloproteins and the induction of oxidation of amino acid side chains [83].

Another possible mechanism could be the interference of metal ions with the achievement of protein concentrations necessary for the autocatalytic aggregation process [84]. In fact, the high concentration of molecules in the intracellular environment, known as “macromolecular crowding”, can influence various biological processes, such as protein-molecules interactions and protein shape, folding [85], oligomerization, and aggregation [86,87].

Pesticides have also raised many concerns regarding their impact on neurodegenerative disorders [88].

In the late 1990s, during the BSE epidemic in the United Kingdom, commonly known as the “mad-cow outbreak”, in an attempt to solve the mystery of its origin, some observations led scientists to suspect a potential role of pesticides. In particular, the use of organophosphate, which determines an involvement of the entire physiology of the animal, attracted attention [89]. Although the most accepted theory of the origin of “mad cow” is prion transmission linked to the use of scrapie-contaminated meat and bone meal from diseased sheep, co-contamination through pesticides could potentially have triggered the pathological process. Following these observations, other in vitro studies on cell cultures have shown that the native prion protein was expressed more in the presence of organophosphate and, therefore, more likely to interact with its misfolded counterpart in order to initiate the pathological process [90]. Furthermore, it has been shown that even the use of low doses of a bioherbicide is able to favor the aggregation and propagation of prions in vivo [91].

In the case of PD, other pesticides, such as paraquat and rotenone, have been blamed for the increased risk of developing the disease in agricultural workers and people living in contaminated rural areas [92]. In support of this evidence, experimental studies reported that the administration of rotenone can trigger a Parkinsonian’s syndrome in animal models [93]. Although the mechanisms of interactions are still a matter of debate, pesticides have also been shown to accelerate protein aggregation under conditions of molecular crowding [94].

### 4.5. Role of Gut Microbiota

The intestine is most likely the origin of the pathology associated with both prions and α-synuclein, representing the critical environment for the formation of misfolded proteins.

Indeed, many of the hypothesized environmental factors continuously and simultaneously come into contact with the intestinal mucosa, which is intimately associated with the ENS, and can act as an initial trigger for the involvement in the *misfolding* process of native proteins. Metals, xenobiotics, pesticides, pathogens, and toxins are among other possible co-factors that have access to the intestine through the oral route and food intake. Interestingly, these same elements are considered among the factors that can have a strong impact on the balance between beneficial and harmful microbes that inhabit the gut, which represent the consortium of approximately 10^14^–10^15^ bacteria present in the gut microbiota [95,96,97]. In fact, even if the gut microbiota has been defined by some authors as “the forgotten organ” [98], in recent years it has regained its spotlight, as it can orchestrate a wide range of physiological functions, well beyond the intestinal one: from metabolism [99], neurogenesis [100], protection against pathogens [101], to development and regulation of host immunity and the preservation of the integrity of the intestinal barrier [102].

Over recent decades, scientists have introduced the revolutionary concept that gut microbiota can influence the function and behavior of its host’s brain by establishing a bidirectional communication with the brain known as the “gut–brain axis” [103]. The gut–brain axis has been defined as a paradigm shift in neuroscience [104], in which gut microbes and their signaling molecules communicate with the brain through neural, endocrine, immune, and metabolic pathways [105,106,107].

Emerging evidence suggests that an alteration in the composition of gut microbiota, called dysbiosis, can have a profound impact on the gut–brain crosstalk with pathophysiological consequences such as gut and brain dysfunctions and inflammation, thus increasing host susceptibility to gastrointestinal and neurological disorders.

Although PD is primarily considered a neurological disorder, gastrointestinal symptoms, including constipation, prolonged bowel transit time, dysphagia and pathophysiological changes in the intestinal barrier, appear many years earlier. These gut impairments might be caused by an alteration of the neurotransmission pathways between the central level and the periphery [108,109,110,111].

Given the involvement of the gastrointestinal tract in PD, many scientists have investigated the potential role of gut microbiota and its interaction with the host, which could become harmful and act as a triggering event in the onset and/or progression of the disease. 

Evidence emerging from an animal model of PD suggests that the gut microbiota can promote both neuroinflammation, a common feature of PD [112], and α-synuclein aggregation, and impair motor function through the activation of microglia [113].

Furthermore, pioneering and ongoing human studies have correlated the alteration in the composition of gut microbiota with PD [114,115,116,117,118,119,120,121,122,123]. Although a general agreement on which microbial species are mainly involved has not yet been reached, the most relevant dysbiotic pattern in patients with PD is characterized by a shift in relative bacterial abundances with a prevalence of pathobionts belonging to the phylum Proteobacteria and to the Enterobacteriaceae family [117,119,120,123]. On the other hand, beneficial symbionts belonging mainly to the Prevotellaceae and Lachospiraceae families, and some genera such as *Blautia*, *Roseburia*, and *Faecalibacterium* are less represented [116,117,120,123]. In fact, a shift of the gut microbial community towards harmful symbionts induces an immune response, which promotes an inflammation state that favors a habitat in which Enterobacteriaceae grow better [124].

As a result, an increase in the abundance of Enterobacteriaceae is associated with an increase in the release of lipopolysaccharide (LPS), a component of the outer membrane of Gram-negative bacteria, which is a potent inducer of inflammatory response, oxidative stress, and is responsible for increased permeability of the intestinal and cerebral barriers [124]. In addition, several studies have shown that an inflammatory environment triggered by LPS and, consequently, an alteration of intestinal mucosa closely related to the ENS, can lead to an increase in the expression and aggregation of α-synuclein in the ENS [110,125,126]. Consistent with these observations, intestinal inflammation has been reported in patients with PD, correlated with an increase in proinflammatory cytokines in the colon, in markers of intestinal permeability and in LPS-binding proteins in plasma [109,110,127,128].

Important members of the Enterobacteriaceae, such as *Escherichia coli,* can produce bacterial amyloid proteins, called curli. Through a cross-seeding mechanism that mimics the host’s protein structures, these amyloid proteins can function directly as a template for the formation of α-synuclein aggregates. Alternatively, with an indirect mechanism, curli can activate the host’s immune response, thus promoting the activation of microglia and astrocyte, and favoring the progression of amyloid aggregates towards the CNS [128].

Interestingly, *curli* fimbriae can be present in the human intestine [129] and their oral administration can induce increased neuronal deposition of α-synuclein in both the gut and brain tissues of aged Fischer 344 rats [130]. Otherwise, a reduction of Prevotellaceae, which are also involved in the production of mucin in the gut mucosal layer, and the Lachnospiraceae members and *Faecalibacterium* can lead to a reduction in the production of short-chain fatty acids (SCFAs).

The SCFAs, such as acetate, propionate, and butyrate, are produced by bacterial fermentation in the colon and represent one of the most important classes of signal molecules in the gut–brain axis. These key molecules, endowed with antioxidative and anti-inflammatory properties, play an important role in orchestrating the function of the enteric nervous system and promoting the integrity of the intestinal mucosal barrier and gastrointestinal motility, and maintaining immune homeostasis. A reduction of these important molecules can have an impact on gastrointestinal functions and contribute to an inflammatory environment by inducing the so-called “leaky gut syndrome”, which appears to be associated with the aforementioned tendency for the formation and propagation of α-synuclein from the gut to the brain [131].

Interestingly, a recent study in an animal model of prion disease reported similar alterations in the gut microbiota with an overgrow of Proteobacteria and a decrease in Prevotellaceae, associated with a reduction in SCFAs concentration [98]. Several studies have suggested that exposure to commensal bacteria and bacteria components, particularly LPS, may promote LPS-prion protein interaction [132], causing an overexpression of normal PrP^C^ in the gut mucosa [133], which could trigger its conversion into aggregate forms. In fact, it has been shown that wherever PrP^C^ is highly expressed, misfolded prions are able to spread exponentially [134,135].

As with Parkinson’s disease, these changes can impact the gut-brain axis by inducing intestinal inflammation, increasing gut and brain–blood barrier (BBB) permeability, activating microglia (one of the earliest signs of prion disease), and facilitating the access to the CNS of neurotoxic substances and/or abnormal prion proteins.

Another fundamental aspect that supports the relationship between gut microbiota-brain axis and PD/prion pathogenesis concerns the effects of antibiotic treatments. Several studies underline that in different experimental models of PD, in addition to the antimicrobial activity, anti-inflammatory, neuroprotective, and anti-aggregation effects have been shown [136]. Similar beneficial effects were observed in terms of increased survival time in hamsters that suffered from prion disease and in patients with sporadic CJD treated with antibiotics [137,138]. The results on the effects of antibiotic therapy in slowing the disease progression further suggest an interesting correlation between gut microbiota and PD/prion pathogenesis. However, the central issue that still remains unclear concerns the cause-and-effect relationship between gut dysbiosis, α-synuclein, and prion diseases.

## 5. Conclusions

Seminal works have tried to answer the question of whether PD can be considered a “prionopathy” because α-synuclein exhibits prion-like characteristics.

The similarities in the pathogenetic characteristics discussed above (e.g., the aggregation path, cell to cell transmission; point of origin, host dissemination) and in the co-factors involved in the misfolding process (e.g., PTMs, lipid interactions, metals ions, pesticides, molecular crowding, and gut microbiota) seem to support this hypothesis.

On the other hand, the substantial transmissible nature of prion disorders has led to define these pathologies as infectious diseases, where the transmission from “reservoir host” to other animal species and to humans has been proved beyond any doubt. However, human transmission of a-synuclein has not yet been established. The lack of such evidence emphasizes the main difference between the pathologies, ultimately suggesting that PD is not to be considered a real “prionopathy”.

However, many similarities remain, and should be considered to better understand the factors underlying the pathogenesis of PD and prion disease: starting, but not limited to, from the central event, which consists in the process of self-aggregation and propagation of misfolded proteins. Other factors, which are closely related to this pathological event and make PD a multifactorial disease, should be taken into consideration.

As the intestine is believed to be a possible site of origin of the disease, it represents the environment in which various cofactors able to promote the protein aggregation process can coexist and act.

This review provides a point of view on the relationship between α-synuclein and the microbiota–gut–brain axis, in which multiple factors potentially implicated in the pathogenetic process of associated disease can coexist: the intestine can be therefore considered a key site for the development of neurodegenerative disorders associated with the misfolding and propagation of prion-like proteins, similar to how prionopathies work.

## Data Availability

Data sharing not applicable.

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
