# Peer review of "Parkinson’s Disease: A Prionopathy?"

_ijms, 2021, doi:10.3390/ijms22158022_

Round 1

Reviewer 1 Report

PUBMED listed over 40 reviews with the search narrowed to keywords parkinson[title] AND synuclein[title] published in 2020 and 2021 alone, so this is yet another review on this subject. This particular review stands out from the crowd by briefly summarizing every aspect of a-synuclein involvement in  PD pathogenesis, whereas other reviews tend to concentrate on the particular aspects of the disease. As such it may be useful for the people just entering the field.

The review may be more comprehensive if a chapter on a-synuclein gene mutations is included. Post-translational modifications could also be briefly discussed.

I noticed some language issues on the first few pages but the whole manuscript should be checked once again for misspelled and misused words.

Some examples:

Line 28 -   composed by  should be composed of

l.45 unfolded proteins, which are characterize by low hydrophobicity

l.47 Whereas the tail consisting of the C-terminal contains the sites...  ( C-terminal end contains the sites)

  1. 56 pioneering not pioneristic

L.63   Kuru who? Kuru is not a person

Author Response

Reviewer 1

PUBMED listed over 40 reviews with the search narrowed to keywords parkinson[title] AND synuclein[title] published in 2020 and 2021 alone, so this is yet another review on this subject. This particular review stands out from the crowd by briefly summarizing every aspect of a-synuclein involvement in PD pathogenesis, whereas other reviews tend to concentrate on the particular aspects of the disease. As such it may be useful for the people just entering the field.

 Point 1: The review may be more comprehensive if a chapter on a-synuclein gene mutations is included. Post-translational modifications could also be briefly discussed.

Response 1: We are extremely grateful to the Reviewer 1 for her/his comments and suggestions. We provided the following chapters in the section “Co-factors involved in the misfolding process” (pages 5-6, lines215-259):

- Gene mutations

Although most human prion diseases and PD are sporadic, 10% - 15% of cases are classified as familial, which indicates the critical role of gene mutations in the pathogenesis of these disorders [65]. Mutations in the prion protein gene (PRNP) are associated with inherited forms of GSS (e.g, P102L, P105L, P105S, A117V, F198S, Q217R), FFI (e.g., D178N) and gCJD (e.g., D178N, V180I, H187R, E200K, R208H, V210I) with autosomal dominant patterns [66]. While fewer gene mutations represent octapeptide expansions or deletions, most prevalent mutations are point mutations (missense mutations) located in the N- or C-terminal region of PRNP gene and they show a complete penetrance. On the contrary, E200K and V210I mutations are associated with a variable penetrance de-pending on age [66, 67]. Likewise, most of inherited forms of PD appear to be caused by point mutations (missense mutations) or multiplications (duplication and triplication) of the α-synuclein gene (SNCA) with incomplete penetrance depending on age [65, 68]. The pathogenic mutations linked to SNCA gene (e.g., A53T, A53E, A30P, E46K, H50Q, and G51D) occur in the N-terminal region of α-synuclein and are related to an early onset and rapid progression of PD. Conversely, mutations of Leucine-rich repeat kinase 2 (LRRK2), which represent an increased risk for the development of inherited PD, are associated with a late-onset of the disease (e.g., G2019S) [68]. Whereas the predominant pathogenic mutations in both SNCA and LRRK2 result in autosomal dominant pattern similarly to inherited forms of human prion diseases, other mutations (e.g., PARK2, PARK7 and PARK6) that involve PRKN, DJ-1 and PINK1 genes result in autosomal recessive patterns [65].

- Post translational modifications

A dysregulation of post-translational modifications (PTMs), due for example to in-flammation conditions, can alter the structure of susceptible proteins, such as PrPC and α-synuclein, promoting their transition into toxic oligomers, fibrils and aggregates [64, 69, 70]. PrPC and α-synuclein undergo to PTMs, such as addition of functional groups and proteolytic cleavage (e.g., glycosylation, acetylation, nitration, oxidation and phosphor-ylation), during and after their biosynthesis. In particular, the common pathological PTMs of both PrPC and α-synuclein proteins include phosphorylation and nitration. Phos-phorylation induce anionic conditions that favor the conformational changes of proteins [64]. While pathological phosphorylation of PrPC mediated by cyclin-dependent kinase 5 involves N-terminal region at amino acid residue Ser-43, the prevalent phosphorylation of α-synuclein involves the C-terminal region at amino acid residue Ser-129 and can be mediated by several kinases (e.g., casein kinase I and II, the G protein-coupled receptor kinases, Leucine-rich repeat kinase 2, and polo-like kinases) [69, 70]. These phosphorylated sites are associated with the self-aggregation of the proteins and the development of proteinopaties in human prion diseases and PD. Regarding PD, it is indeed known that over 90% of α-synuclein in the Lewy bodies is phosphorylated at Ser-129. Moreover, nitration modifications of tyrosine residues, which are induced by the action of oxygen and nitric oxide and their products, involve multiple residues of both PrPC (Y131, Y148, Y152, Y153, Y158, Y221, Y158, Y221, Y227/Y228) and α-synuclein (Y125, Y133 and Y136) and occur predominantly in the C-terminal region of the proteins. Several evidences have demonstrated that nitration increases the aggregation of PrPC and the oligomerization of α-synuclein [69, 70].

Point 2: I noticed some language issues on the first few pages but the whole manuscript should be checked once again for misspelled and misused words.Some examples:

Line 28 -   composed by should be composed of

l.45 unfolded proteins, which are characterize by low hydrophobicity

l.47 Whereas the tail consisting of the C-terminal contains the sites...  (C-terminal end contains

thesites)

1.5 pioneering not pioneristic

L.63   Kuru who? Kuru is not a person

Response 2: We comply with this Reviewer points and we provided to fix the language issues indicated by Reviewer. We also provided an extensive English revision of the manuscript following the Reviewer’s suggestion.

Reviewer 2 Report

In this review manuscript, the authors highlight the similarity of the main pathological characteristics of misfolded α-synuclein and prions in structures, aggregation pathways, cell to cell transmission, origin, and co-factors involved in their misfolding process. At the end, the authors indicate the key differences between the two: prion diseases are infectious diseases and can be transmitted from individuals to individuals while PD is a multifactorial disease and it has no individual transmission to be confirmed.

This review is very interesting and it highlights recent advances in transmission properties of misfolded α-synuclein whose deposition in the brain is the hallmark of Parkinson’s disease. It is comprehensive and well-balanced. Many new and important studies have been cited in this review manuscript. There are only a few issues that the authors may want to address.

  1. As the authors indicate in the manuscript, because α-synuclein has recently been found by several lines of evidence to exhibit prion-like properties, one wants to know whether PD can be considered as a prionopathy, which also reflects in the title of the manuscript. However, the authors do not seem to provide some kind of answers at the end. It will be helpful if the authors can provide a pros and cons list to help readers understand the current views in the field better after this comprehensive review.
  2. It may cause confusion to claim that prion disease “commonly originates in the periphery.” in line 138, page 3. This is because it may apply to certain animal prion diseases such as BSE and CWD but not to most of sporadic and familial prion diseases in humans.
  3. The authors discuss the evidence about the two possible origins of misfolded α-synuclein including “brain first” and “body-first”. However, for prion disease, the authors only discuss the prion dissemination via the peripheral gastrointestinal tract to the CNS. How about prion transmission from the CNS to peripheral tissues and organs?

Author Response

Reviewer 2

In this review manuscript, the authors highlight the similarity of the main pathological characteristics of misfolded α-synuclein and prions in structures, aggregation pathways, cell to cell transmission, origin, and co-factors involved in their misfolding process. At the end, the authors indicate the key differences between the two: prion diseases are infectious diseases and can be transmitted from individuals to individuals while PD is a multifactorial disease and it has no individual transmission to be confirmed.

This review is very interesting and it highlights recent advances in transmission properties of misfolded α-synuclein whose deposition in the brain is the hallmark of Parkinson’s disease. It is comprehensive and well-balanced. Many new and important studies have been cited in this review manuscript. There are only a few issues that the authors may want to address.

Point 1: As the authors indicate in the manuscript, because α-synuclein has recently been found by several lines of evidence to exhibit prion-like properties, one wants to know whether PD can be considered as a prionopathy, which also reflects in the title of the manuscript. However, the authors do not seem to provide some kind of answers at the end. It will be helpful if the authors can provide a pros and cons list to help readers understand the current views in the field better after this comprehensive review.

Response 1: We are grateful to the Reviewer 2 for your comments and suggestions. We agree and provide the following paragraph in the Conclusion section (page 9, lines 417-428).

The similarities in the pathogenetic characteristics discussed above (e.g., the aggregation path, cell to cell transmission; point of origin, host dissemination) and in the co-factors involved in the misfolding process (e.g., PTMs, lipid interactions, metals ions, pesticides, molecular crowding, gut microbiota) seem to support this hypothesis.

On the other hand, the substantial transmissible nature of prion disorders has led to define these pathologies as infectious diseases, where the transmission from "reservoir host" to other animal species and to humans has been proved beyond any doubt. However, human transmission of a-synuclein has not yet been established. The lack of such evidence emphasizes the main difference between the pathologies, ultimately suggesting that PD is not to be considered a real "prionopathy".

Point 2: It may cause confusion to claim that prion disease “commonly originates in the periphery.” in line 138, page 3. This is because it may apply to certain animal prion diseases such as BSE and CWD but not to most of sporadic and familial prion diseases in humans.

Response 2: We comply with this Reviewer point and we provided the following integration (page 3, lines 140-1419):

“Although prion diseases are mainly limited to the central nervous system (CNS), except for sporadic and familial forms, most of them originate in the periphery”

Point 3: The authors discuss the evidence about the two possible origins of misfolded α-synuclein including “brain first” and “body-first”. However, for prion disease, the authors only discuss the prion dissemination via the peripheral gastrointestinal tract to the CNS. How about prion transmission from the CNS to peripheral tissues and organs?

Response 3: We agree. We provided the following integration in the paragraph “point of origin and host invasion” (page 4, lines 154-161):

On the contrary, prion dissemination from the CNS to peripheral tissues and organs occurs through anterograde spreading along nerve fibers, peripheral synapses (e.g., neuro-muscular junction) peripheral tissues (i.e., muscle cells, mucosa), lymph and blood [50]. Interestingly, an example of prion dissemination to the peripheral mucosa is documented in the olfactory system. Here prions spread along synapses within olfactory neurons via the olfactory and vomeronasal cranial nerves and reach olfactory sensory mucosa with subsequent release of prions into the nasal cavity [50-52]”
